

# Human and conservation factors affect spatial variation of reef fish assemblages in Colombian Pacific reefs

Juan P. Quimbayo[1], Luis Chasqui[2], Natalia Rincón-Díaz[2,3], Adriana Alzate[4,5] and Fernando A. Zapata[6]

[1] BioScales Lab, Department of Biology, University of Miami, Miami, FL, United States of America
[2] Instituto de Investigaciones Marinas y Costeras INVEMAR, Santa Marta, Magdalena, Colombia
[3] Academic Area of Biology and Environmental Sciences, University of Bogotá Jorge Tadeo Lozano, Santa Marta, Magdalena, Colombia
[4] Naturalis Biodiversity Center, Leiden, the Netherlands
[5] Aquaculture and Fisheries Group, Wageningen University and Research, Wageningen, the Netherlands
[6] Departamento de Biología, Universidad del Valle, Cali, Valle del Cauca, Colombia

Corresponding author
Juan P. Quimbayo,
quimbayo.j.p@gmail.com

## ABSTRACT

Humans have both negative and positive impacts on marine communities: human everyday activities can degrade ecosystems, while conservation efforts can support their protection and recovery. Using an empirical database of fish assemblages compiled from 393 underwater visual censuses along the Colombian Pacific Coast, we assessed spatial variation in these assemblages and investigated whether they are shaped by human pressures, such as number of fishers and proximity to markets, as well as conservation measures, including protection status and the age of the marine protected areas. Our study reveals that remote locations have a higher fish density and biomass than those near the coast. We found that grunts (Haemulidae) were the most species-rich family and contributed the most to fish density and biomass. Piscivores were the trophic group most affected by human factors, showing lower species richness, density, and biomass in coastal locations. In contrast, other trophic groups did not show a negative response to human factors across locations. We did not observe an effect of human and conservation factors on the total species richness. To evaluate the potential influence of Malpelo island, the only oceanic location in the dataset, we built two models, one with and one without this site. The results were consistent across both models, indicating that including or excluding Malpelo did not alter overall patterns of species richness. However, market distance negatively influenced the average fish density and biomass when all locations were included. Our results provide the first quantitative assessment of fish assemblages across the Colombian Pacific Coast, enabling future comparison and enhancing our understanding of the effects of human and conservation activities on the patterns of species richness, density, and biomass of reef fishes in the Eastern Tropical Pacific.

## INTRODUCTION

Human activities, such as fishing, species introductions, and the destruction of natural habitats, significantly impact current biodiversity patterns (*Halpern et al., 2008*; *Mora et al., 2011*). In marine habitats, fishing substantially contributes to declines in the abundance and biomass of several fish species (*Mumby et al., 2012*; *Graham et al., 2017*). For instance, over-fishing has promoted local extinctions of top predators such as sharks in oceanic islands (*Luiz & Edwards, 2011*), with consequences for trophic cascades resulting in a drastic reduction of coral cover in these locations (*Sandin et al., 2008*). Human pressures on ecosystems can vary widely on spatial and temporal scales (*Ruppert et al., 2018*), depending on the social, cultural, and economic conditions (*Halpern et al., 2008*), as well as the proximity and density of human populations (*Cinner et al., 2018*). While human impacts on marine biodiversity tend to be negative, conservation efforts, such as marine protected areas (MPAs), aimed at mitigating negative effects can be conceptualized as potentially positive human impacts (*Topor et al., 2019*; *Davidson & Dulvy, 2017*). Evidence supports that MPAs can maintain a high abundance and biomass of fish compared to fished areas, resulting in marine communities that are highly resilient and resistant to human impacts (*Aburto-Oropeza et al., 2011*; *Topor et al., 2019*). Furthermore, MPAs can act as "sources" that promote biodiversity and sustainable fisheries through spillover effects (*Fountoura et al., 2022*). Thus, understanding the impact of human and conservation factors' influence on fish assemblages is crucial for informing the development of more effective conservation and restoration strategies for coastal ecosystems.

Reef fish constitute the most diverse vertebrate group in marine habitats and serve as a primary source of animal protein and economic sustenance for many coastal human populations (*Pinca et al., 2012*). Consequently, many fish species are subject to overfishing, and some are on the brink of population collapse (*Newton et al., 2007*). Species that show strong declines in abundance and biomass include predators and large herbivores such as groupers and snappers, surgeonfishes, and parrotfishes, which are fishing targets of many fishing methods (*Ceretta et al., 2020*; *Giglio et al., 2020*; *Roos et al., 2020*). The overexploitation of these species (*i.e.,* top predators and large herbivores) can impact top-down control in ecosystems, potentially leading to alterations in ecosystem functions like nutrient cycling and species regulation (*D'agata et al., 2014*; *Brandl et al., 2019*; *Mumby et al., 2012*; *Ruppert et al., 2013*; *Barneche et al., 2019*). Despite global consensus on the impacts of human activities on fish assemblages in coastal areas (*Brewer et al., 2013*; *Cinner et al., 2018*; *Cinner et al., 2013*), these can vary among locations or regions, thus studies at regional scales can bring additional insights about how human factors (*e.g.,* number of fishermen or market distance) and conservation initiatives influence fish communities.

In regions or countries where there is a scarcity of socioeconomic data and limited government funding for conservation efforts, regional biodiversity assessments often lack baseline information regarding the impact of human activities on fish assemblages (*Pauly, Watson & Alder, 2005*; *Béné, Hersoug & Allison, 2010*). The problem with these knowledge gaps is that they preclude the identification of critical shifts in fish assemblages caused by current fishing practices, thus hindering the implementation of necessary

conservation measures (*Pauly, Watson & Alder, 2005*; *Evans, Cherrett & Pemsl, 2011*). Colombia, classified as a developing country (*Arango-Aramburo et al., 2017*), has two coastal regions—a Caribbean coast and a Pacific coast (*Correa & Morton, 2010*). Historically, the Pacific coast of Colombia has been politically and economically marginalized, resulting in one of the country's most isolated and economically disadvantaged regions, characterized by low human population density (*Castiblanco, Etter & Ramirez, 2015*). Furthermore, a continuous armed conflict spanning more than 50 years has exacerbated the isolation and marginalization of the region, severely limiting the assessment of fish assemblages—a situation that accounts for the small number of studies in this area (*e.g.*, *Gomez & Vieira, 1996*; *Zapata & Morales, 1997*; *Tobón-López, Rubio & Giraldo, 2008*; *Palacios & Zapata, 2014*; *Quimbayo et al., 2017b*; *Chasqui, 2020*).

Here, we have compiled an empirical database of 393 underwater visual censuses conducted at seven locations along the Colombian Pacific Coast with two goals in mind: (1) examine the variation in species richness, density, and biomass of fish assemblages along the coast and among different trophic groups; and (2) test whether human and conservation factors, such as the number of fishermen, market distance, conservation status, and Marine Protected Area age, have an impact on species richness, density and biomass of fish assemblages along the coast and among different trophic groups. This study provides the first quantitative assessment of fish assemblages across the Colombian Pacific Coast, enabling future comparison and enhancing our understanding of the effects of human and conservation activities on the patterns of species richness, density, and biomass of reef fishes in the Eastern Tropical Pacific.

## MATERIALS AND METHODS

### Study area

The Colombian Pacific Coast (CPC) is on the southern portion of the Eastern Tropical Pacific region (*Robertson & Cramer, 2009*). This coast has an extensive coastline (1,300 km) and exhibits a variety of environments, such as mangroves, rocky reefs, and coral reefs (*Zapata & Vargas-Ángel, 2003*; *Cantera & Londono-Cruz, 2011*) (Fig. 1A). The mangrove forest along the Colombian Pacific Coast is the most extense in the country with 132,099 hectares, characterized by high primary productivity but low diversity (*Mejía-Rentería et al., 2018*). Reefs in the region consist mainly of rocky reefs, which are submerged rock outcrops with varying relief, creating refuges for small fishes and increasing surface areas for the colonization of algae and invertebrates. Specifically, coral reefs in the CPC are modest and exhibit low taxonomic diversity (approximately 21 coral species), in part due to marginal oceanographic conditions for reef development, including high precipitation, increased sedimentation, and occasional intense El Niño events that elevate sea surface temperatures and lead to coral bleaching and mortality (*Zapata & Vargas-Ángel, 2003*). Additionally, the seasonal latitudinal displacement of the intertropical convergence zone affects winds and currents in the Gulf of Panama and Colombia and causes seasonal variation in nutrient availability due to upwelling (*Cantera & Londono-Cruz, 2011*). Numerous traditional fishing communities and small cities that depend mainly on fishing activities to obtain

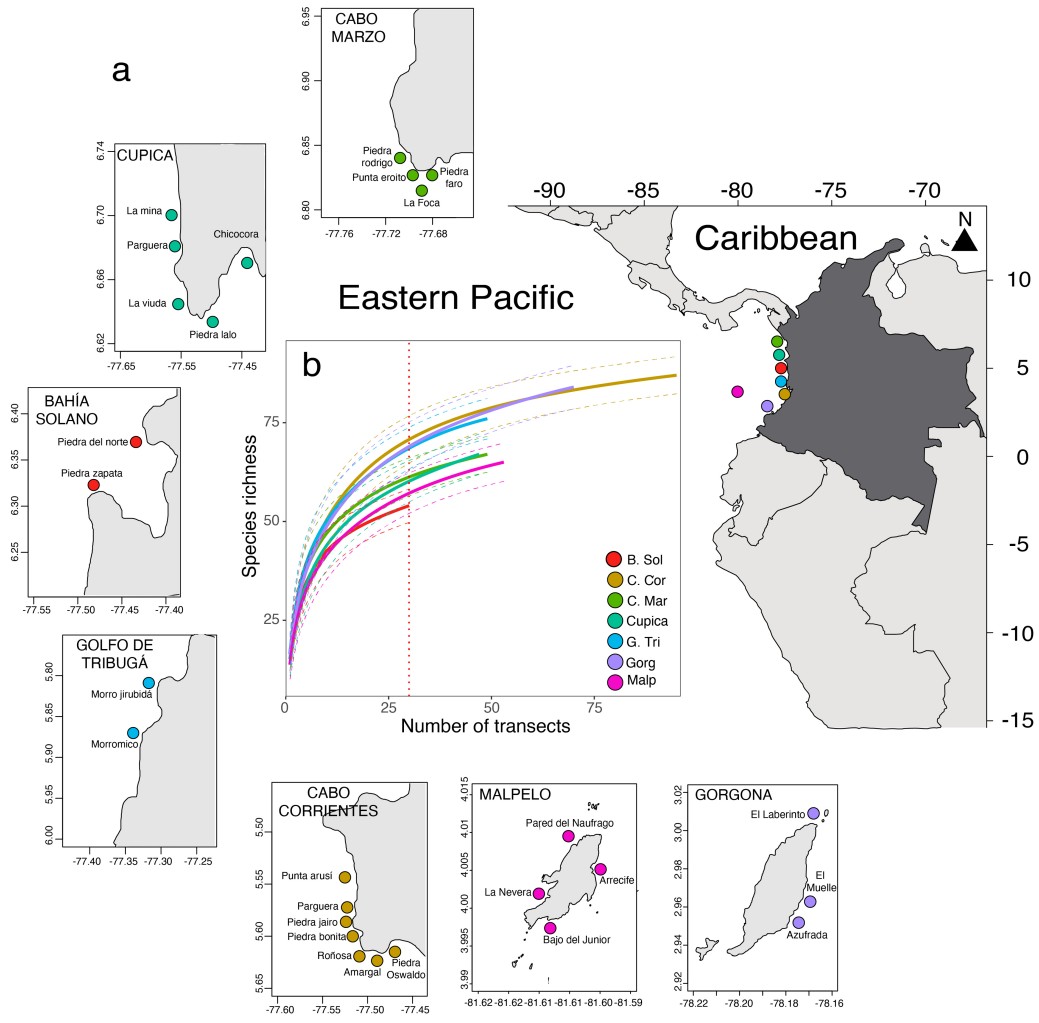

**Figure 1** Seven locations and 28 sites sampled along of the Colombian Pacific Coast (A). Comparison of fish species richness observed in 393 underwater visual censuses in the seven locations sampled (B).

food provision and economic resources are situated along the Colombian Pacific Coast (*Castellanos-Galindo & Zapata, 2018*; *Herrón et al., 2018*; *Herrón et al., 2019*).

Between 2006 and 2016 during the rainy season, we sampled over 28 sites across seven locations on the Colombian Pacific Coast (Table S1): Cabo Marzo, Cupica, Bahía Solano, Golfo de Tribugá, Cabo Corrientes, Gorgona, and Malpelo (Fig. 1A). These locations vary in their isolation from the mainland, conservation status, and human pressures. For instance, Cabo Marzo is the northernmost location, while Gorgona is the southernmost, and Malpelo is the only oceanic island, located 380 km from the coast (Table S1). All these locations are mainly rocky reef areas with some coral outcrops and overgrowth (*Zapata & Vargas-Ángel, 2003*). Gorgona and Malpelo are the oldest protected areas (39 and 29 years, respectively), followed by Golfo de Tribugá and Cabo Corrientes, each with 10 years of protection. In contrast, Cabo Marzo, Cupica, and Bahía Solano, are unprotected (*Guzman*

*et al., 2023*). Some of these locations are remote (*i.e.,* Malpelo Island) and historically were colonized by Afro-Colombian communities, which use artisanal fishing as an alternative activity to obtain food (*Cobos-Otálora et al., 2012*; *Castellanos-Galindo & Zapata, 2018*).

## Fish surveys

We carried out 393 underwater visual censuses (UVCs) of fish assemblages distributed across the Colombian Pacific Coast (Fig. 1; Table S1). In each UVC, a diver identified, counted, and estimated total length in cm ($T_L$) of all species observed within a belt transect. The area per belt transect varied between 40 and 60 m$^2$ (Table S1). We estimated biomass (fish weight in grams) of each individual fish (W) using the allometric length-weight conversion:

$$W = a \times T_L^b, \tag{1}$$

where $T_L$ is the total length and parameters *a* and *b* are allometric species-specific constants (*Granfeldt, 1979*). Specific length-weight parameters were obtained for nearly all species from *Quimbayo et al. (2021)*. Further, we classified fish species observed in the UVCs into seven trophic groups according to *Quimbayo et al. (2021)*: herbivores-detritivores (feed upon turf and filamentous algae and/or detritus), macroalgae-feeders (feed on large fleshy algae and/or seagrass), sessile invertebrate feeders (feed on corals, sponges, ascidians), mobile invertebrate feeders (feed on benthic prey, such as crabs and mobile mollusks), planktivores (feed on small organisms in the water column), piscivores (feed on fish and cephalopods) or omnivores (feed on algae, detritus, and animal material). Fish data from Gorgona were extracted from *Alzate (2022)*. All the shark and ray species were excluded from the analysis since they may disproportionally increase the biomass values, especially in belt transects with small areas (*Ward-Paige, Flemming & Lotze, 2010*).

## Human and conservation factors

We considered the number of fishermen and market distance as human factors potentially having negative impacts. The number of fishermen was extracted from national reports (*Zuluaga et al., 2009*; *Cobos-Otálora et al., 2012*; *Velandia & Diaz, 2016*). Market distance was estimated linearly using the site coordinates (Table S1) to identify the market closest to each location since none of the reefs visited were inside mangrove areas. Additionally, we compiled information on the national protection status of each location based on Colombia's National Registry of Protected Areas—RUNAP (https://runap.parquesnacionales.gov.co/). Lastly, we extracted the Marine Protected Area age from *Guzman et al. (2023)*.

## Data analyses

Due to variation in transect area and sampling effort (number of transects, see Table S1) per site, we implemented a standardized set of steps to adjust the sampling effort for each site. Initially, we randomly selected from each site the number of underwater visual census (UVCs) samples necessary to achieve 65% completeness of species richness. This completeness criterion is based on accumulation curves, which utilize Michaelis–Menten functions to estimate asymptotic species richness based on all available samples from

each site (*Chao & Jost, 2012*; *Maureaud et al., 2020*; *Chao et al., 2020*). We considered 65% completeness because this is the maximum percentage that can be extracted from all sites given the sampling effort. After determining the required number of transects to achieve 65% completeness per site, we estimated fish species richness, density, and biomass per site and trophic group. Given that multiple combinations of UVCs could fulfill this percentage per site, we repeated this procedure 100 times to generate a distribution of estimates for species richness, density, and biomass. Subsequently, we utilized all repetitions to calculate mean species richness, density, and biomass per site and trophic group. This method is analogous to sample-based rarefaction curves, allowing for comparisons between fish assemblage metrics (*Gotelli & Colwell, 2001*), and has been utilized in other local and regional comparisons (*Dubuc et al., 2023*; *Quimbayo et al., 2019*).

To evaluate the differences in fish species richness observed per location, we built rarefaction curves using the *accumcomp* R function from the 'BiodiversityR' package (*Kindt & Coe, 2005*). To compare fish density and biomass among locations, we used two generalized linear models (GLMs) with Gamma distribution and log-link function. Subsequently, we tested for significant differences in fish density and biomass among locations using the *glht* function from the R package 'multcomp' (*Hothorn, Bretz & Peter, 2008*). Additionally, to identify the major contribution to 70% of total dissimilarity observed among locations in terms of density and biomass, we used the similarity percentage (SIMPER) analysis with 999 permutations. We used the *simper* function from the R package 'vegan' (*Oksanen et al., 2015*). Since species composition can vary depending on the combination of transects, we calculated a mean-based similarity percentage estimated in each permutation. We then ranked the species based on the mean value and extracted those species that contributed to the accumulative dissimilarity.

To examine potential collinearity among the negative human impacts (number of fishermen, market distance) and conservation (conservation status and Marine Protected Area age) factors, we used a Pearson's correlation coefficient $r < \pm 0.70$ as cut-off value for retaining factors in the models since values below this threshold in practice are unlikely to involve multicollinearity in models (*Dormann et al., 2013*). We scaled all predictors to a mean of zero and a standard deviation of one to enable the direct comparison among effect sizes. Additionally, we tested for potential multi-collinearity among all predictor variables using the variance inflation factor (VIF) with the *vif* function from the R package 'car' (*Fox & Weisberg, 2019*). We considered that predictors were not correlated with each other with *VIF* values <3 as a cut-off (*Dormann et al., 2013*). We observed a strong correlation between conservation status and marine protected area age ($r = 0.96$, VIF = 17.51). Thus, we did not consider the MPA age in our analysis (Fig. S1, Table 1).

To explore the effect of the fixed factors (number of fishermen, market distance, and conservation status) on fish assemblage metrics (species richness, density, and biomass), we used generalized linear mixed models (GLMMs) with a log link function. We considered a Poisson error distribution for the model explaining species richness and a Gamma error distribution for the model examining fish density and biomass. As UVCs are nested within locations, and locations are nested within years, we included locations and years as random factors. Considering that Malpelo is the only oceanic island, we also re-analyzed
**Table 1  Results of general linear mixed models.** Effects of human and conservation factors on fish assemblage metrics. Significant values in bold. Values between brackets represents the 95% confident intervals.

|  | Species richness | Fish density | Fish biomass | VIF |
|---|---|---|---|---|
| Number of fishermen | −0.02 [−0.11, 0.07] | 0.27 [−0.14, 0.68] | 0.26 [−0.32 0.85] | 2.76 |
|  | $t$-value = −0.50 | $t$-value = 1.39 | $t$-value = 0.94 |  |
|  | $p$-value = 0.63 | $p$-value = 0.18 | $p$-value = 0.36 |  |
| Market distance | −0.07 [−0.16, 0.02] | 0.42 [0.03, 0.84] | 0.57 [−0.05, 1.19] | 2.90 |
|  | $t$-value = −1.63 | $t$-value = 2.26 | $t$-value = 1.90 |  |
|  | $p$-value = 0.12 | $p$-value = **0.008** | $p$-value = **0.031** |  |
| Protection status | 0.03 [−0.03, 0.09] | 0.53 [−0.05, 1.11] | −0.07 [−0.49, 0.34] | 1.38 |
|  | $t$-value = 0.93 | $t$-value = 1.90 | $t$-value = −0.37 |  |
|  | $p$-value = 0.36 | $p$-value = 0.07 | $p$-value = 0.71 |  |
| Number observations | 28 | 28 | 28 |  |
| $R^2$ | −38.92 | −2.25 | −1.23 |  |

the effect of potentially negative human factors and conservation status excluding this location. To include model selection uncertainty (*i.e.,* models with different fixed factors can have similar AICs scores) in the estimation precision of the parameters, we used a model averaging approach. Multimodel inference produced model-averaged (based on AICc) parameter estimates and unconditional standard errors (Adjusted SE) using the *model.avg* R function from the 'MuMIn' package (*Bartoń, 2025*).

To examine how the number of fishermen, market distance, and protection status influence species richness, density, and biomass of fish assemblages in different trophic groups, we used three PERMANOVA analyses using the *adonis2* R function from the 'vegan' package (*Oksanen et al., 2015*). The statistical significance of the PERMANOVA was tested using 999 permutations to reduce type II error (conditional) sums of squares (*Anderson, Gorley & Clarke, 2008*). Lastly, we used non-metric multidimensional scaling (*nMDS*) over Bray-Curtis dissimilarity matrices to represent the variation in species richness, density and biomass in each trophic group, given the effects of human factors and protection status. We used *nMDS* and *envfit* R functions from the 'vegan' package (*Oksanen et al., 2015*). All figures and statistical analyses were performed in the R software, version 4.0.2 (*R Core Team, 2020*).

## RESULTS

### Spatial variation of fish assemblage metrics

We observed a total of 134 fish species belonging to 45 families across seven locations along the Colombian Pacific Coast. The fish density was 4.99 ± 4.56 ind m$^{-2}$ (mean ± SD), and fish biomass was 295.13 ± 460.4 g m$^{-2}$. Per transect, total fish richness ranged from five to 32 species, total fish abundance ranged from 1.30 to 28.97 ind m$^{-2}$, and total fish biomass ranged from 80.43 to 1,710.0 g m$^{-2}$. Golfo de Tribugá, Cabo Corrientes, and Gorgona Island showed the highest species richness, while Bahía Solano exhibited the lowest richness (Fig. 1B). Fish density varied among locations (Fig. 2A; Table S2).

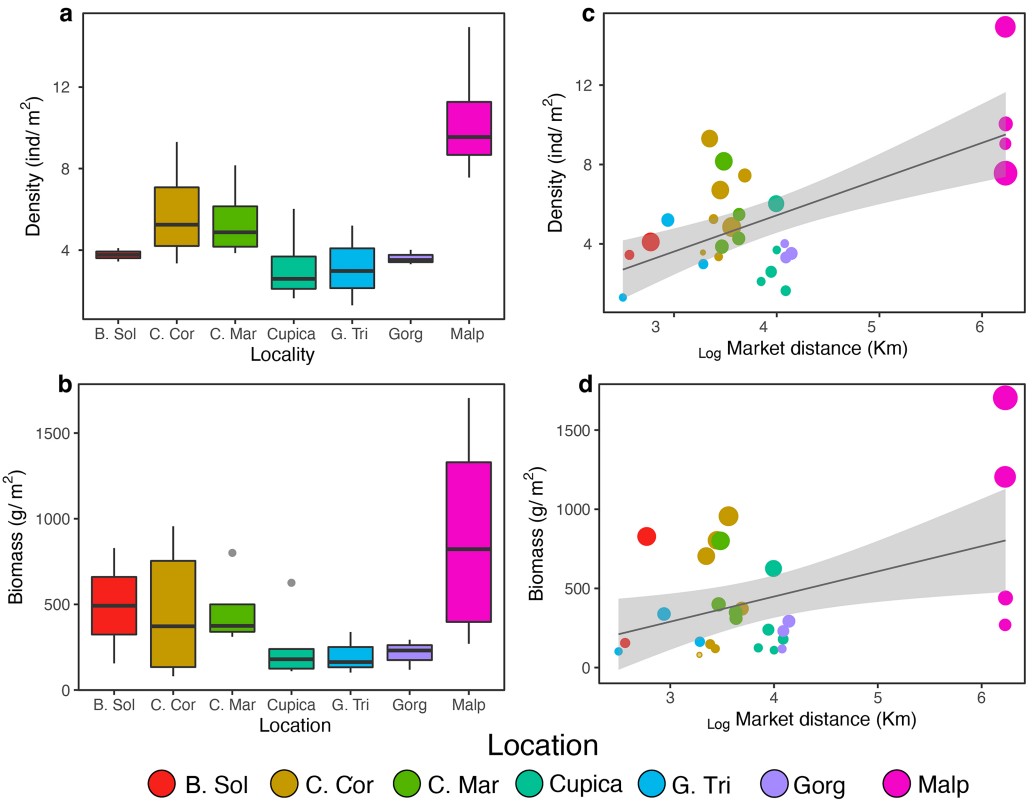

**Figure 2** Comparison density (A) and biomass (B) per location and effect of market distance on density (C) and biomass (D) found from general linear models. Locations: Bahía Solano (B. Sol), Cabo Corrientes (C. Cor), Cabo Marzo (C. Mar), Cupica (Cupica), Golfo de Tribugá (G. Tri), Gorgona (Gorg), and Malpelo (Malp).

Malpelo exhibited the highest fish mean density ($9.59 \pm 4.94$ ind m$^{-2}$), while Cupica had the lowest ($3.41 \pm 2.49$ ind m$^{-2}$ Fig. 2A). Species that contributed the most to overall dissimilarity in terms of density among the locations were the Panamic soldierfish (*Myripristis leiognathus*), Orangeside triggerfish (*Sufflamen verres*), Yellow surgeonfish (*Acanthurus xanthopterus*), and Brassy chub (*Kyphosus vaigiensis*) (Table S4). Similarly, fish biomass also varied among locations (Fig. 2B; Table S2) with Malpelo showing the highest fish mean biomass $798.04 \pm 817.14$ g m$^{-2}$, and Golfo de Tribugá the lowest $141.28 \pm 138.05$ g m$^{-2}$ (Fig. 2B). The primary species that contributed to the dissimilarity among locations in terms of biomass were Mackerel scad (*Decapterus macarellus*), Ember parrotfish (*Scarus rubroviolaceus*), and Pacific dog snapper (*Lutjanus novemfasciatus*) (Table S4).

Groupers (Epinephelidae), wrasses (Labridae), and damselfishes (Pomacentridae) were the most species-rich families observed per transect (>2 species each, Fig. 3A). Cardinalfishes (Apogonidae; $1.31 \pm 0.58$ ind m$^{-2}$), grunts (Haemulidae; $1.28 \pm 1.15$ ind m$^{-2}$), and damselfishes (Pomacentridae; $1.07 \pm 0.91$ ind m$^{-2}$) were on average the most abundant families per transect (Fig. 3B), whereas jacks (Carangidae; $178.64 \pm 348.47$ g m$^{-2}$), grunts

(Haemulidae; 133.05 $\pm$ 218.43 g m$^{-2}$), and snappers (Lutjanidae; 131.53 $\pm$ 258.56 g m$^{-2}$) contributed on average the highest biomass per transect (Fig. 3C). Mobile invertebrate feeders were the trophic group with the highest average species richness per transect at all locations (7.74 $\pm$ 1.51), whereas macroalgae-feeders were the least species-rich group with an average of only one species per transect at all locations (Fig. 3D). Planktivores had the highest average fish density per transect at all locations (2.69 $\pm$ 2.12 ind m$^{-2}$) whereas omnivores had the lowest (0.05 $\pm$ 0.10 ind m$^{-2}$; Fig. 3E). Mobile invertebrate feeders were the trophic group with the highest fish biomass (165.17 $\pm$ 164.21 g m$^{-2}$; Fig. 3F). The proportions of trophic groups in each assemblage varied among locations and depending on the metric. Piscivores represented the highest proportion of total fish richness at Malpelo (20%), whereas planktivores did so at Gorgona (21%) and sessile invertebrate feeders at Golfo de Tribugá and Cabo Marzo (20% each, Fig. 3G). Considering fish density, each trophic group also differed among locations. For example, Malpelo had a higher proportion of piscivores (39%), whereas Cabo Marzo had a higher proportion of herbivores-detritivores (35%), and Cupica of omnivores (42%; Fig. 2H). In terms of fish biomass, piscivores contributed a greater proportion at Malpelo (57%), while planktivores did so at Bahía Solano (55%), and omnivores at Cupica (73%; Fig. 2I).

## Effect of human and conservation factors on fish assemblages

We did not find a significant correlation between average species richness and human and conservation factors (*i.e.,* market distance, the number of fishermen, or the protection status; Table 1 and Table S5) across all locations. However, the average fish density and biomass increased with distance from fish markets (Table 1; Figs. 2C, 2D). After excluding Malpelo Island from GLMMs, we did not observe any effect of human and conservation factors on average species richness, fish density, and biomass (Table S3).

The number of species (*i.e.,* species richness) per trophic group across all locations was influenced by the number of fishermen, market distance, and protection status (Table 2). The number of fishermen had the opposite effect on species richness compared to market distance and protection status. Locations within MPAs such as Gorgona and Malpelo showed segregation somewhat distinct from others, aligning with expectations along this axis (Fig. 4A). Similar effects were observed on the nMDS biplot for fish density (Fig. 4B), and partially for fish biomass since only Malpelo showed the same pattern between the MPAs (Fig. 4C), although fish density and biomass within trophic groups across locations were solely influenced by market distance (Table 2). We also observed more species of piscivores in locations with some protection status, whereas sessile invertebrate feeders were more common in locations with a high number of fishermen (Fig. 4A). Omnivores and macroalgae-feeders densities showed an opposite direction than protection status and market distance (Fig. 4B). Fish biomass of piscivores followed the same direction as protection status, indicating that locations with high protection status had the highest biomass of piscivores, whereas herbivores-detritivores, planktivores, and mobile invertebrate feeders varied in opposite direction to market distance (Fig. 4C).

After excluding Malpelo Island from PERMANOVAs, we did not observe any effect of the number of fishermen and protection status on species richness and fish density per trophic

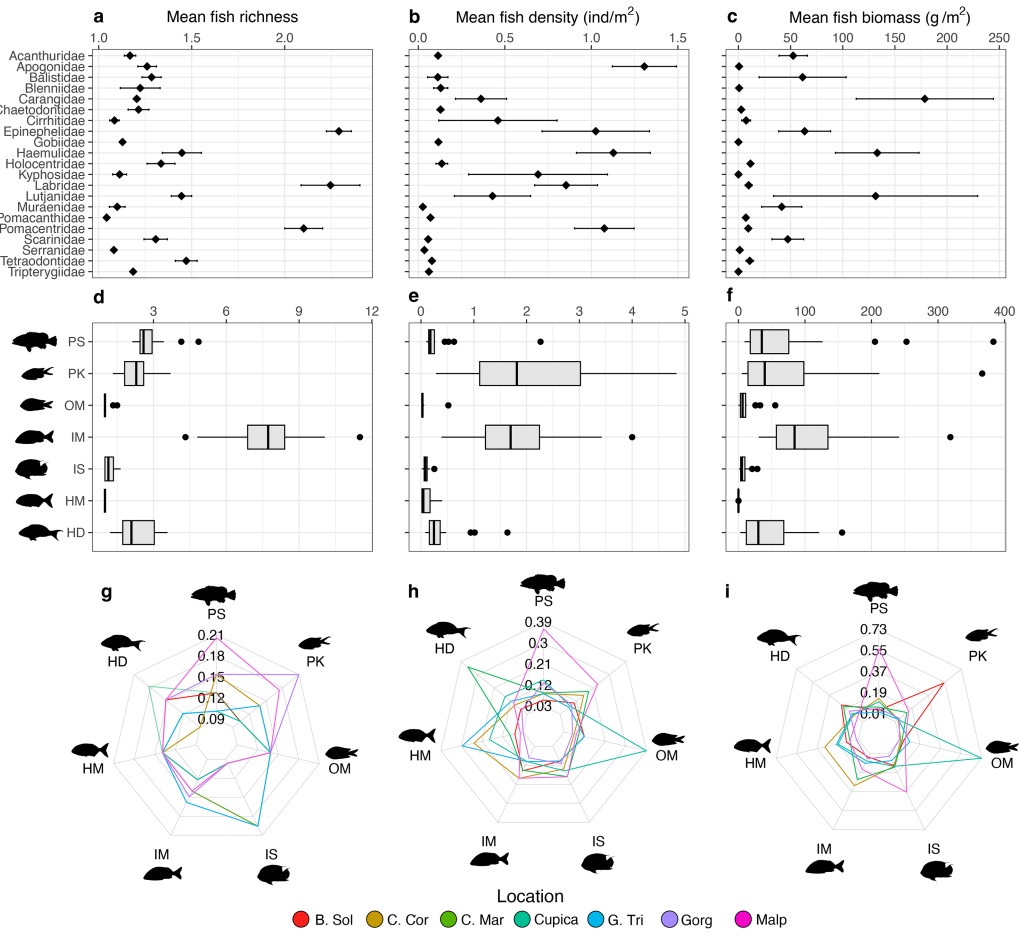

**Figure 3** **Comparisons of fish assemblage metrics per family (top row) and trophic groups (middle and bottom rows).** The metrics include (A) mean species richness, (B) mean density, and (C) mean biomass per family, and (D) mean species richness, (E) mean density, and (F) mean biomass per trophic group. Each heptagon in the bottom row shows the proportional contribution made by each trophic group to the total value observed in (G) mean species richness, (H) mean density, and (I) mean biomass. Trophic groups are: herbivores-detritivores (HD), macroalgae-feeders (HM), sessile invertebrate feeders (IS), mobile invertebrate feeders (IM), planktivores (PK), piscivores (PS), and omnivores (OM).

group (Table S6). However, the effect of market distance on fish biomass per trophic group was still recovered (Table S6). We observed in the nMDS that species richness, density, and biomass of macroalgae-feeders always negatively correlated with market distance (Figs. 4D–4F). On the other hand, species richness and density of sessile invertebrate feeders were negatively related with market distance (Figs. 4D, 4E). Other trophic groups in terms of species richness, density, and biomass were aggregated in the middle of the nMDS biplot (Figs. 4D–4F).

## DISCUSSION

This study is the first comprehensive quantitative regional assessment of reef fish assemblages along the Colombian Pacific Coast (CPC), a region that has been significantly

**Table 2  Results PERMANOVA tests.** Effects of human factors on species richness, density, and biomass observed along the Colombian Pacific Coast. Significant values in bold.

| Fish metric | Factors | df | $R^2$ | $F$-value | $p$-value |
|---|---|---|---|---|---|
| Species richness | Number of fishermen | 1 | 0.09 | 3.41 | **0.03** |
| | Market distance | 1 | 0.10 | 3.55 | **0.01** |
| | Protection status | 1 | 0.12 | 4.30 | **0.01** |
| | Residuals | 24 | 0.68 | | |
| Fish density | Number of fishermen | 1 | 0.05 | 1.63 | 0.18 |
| | Market distance | 1 | 0.12 | 3.71 | **0.02** |
| | Protection status | 1 | 0.02 | 0.51 | 0.68 |
| | Residuals | 24 | 0.79 | | |
| Fish biomass | Number of fishermen | 1 | 0.04 | 1.13 | 0.31 |
| | Market distance | 1 | 0.11 | 3.24 | **0.01** |
| | Protection status | 1 | 0.07 | 0.20 | 0.98 |
| | Residuals | 24 | 0.83 | | |

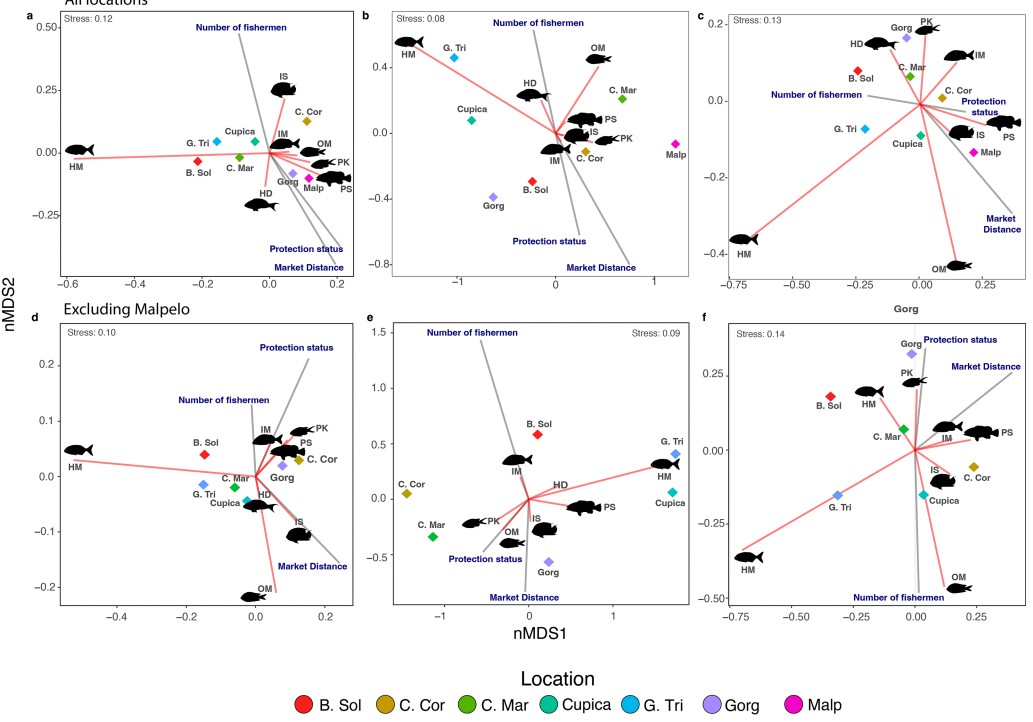

**Figure 4  Non-metric multidimensional scaling of species richness (A, D), density (B, E), and biomass (C, F) observed per trophic group in all locations.** Each diamond represents a location. Locations: Bahía Solano (B. Sol), Cabo Corrientes (C. Cor), Cabo Marzo (C. Mar), Cupica (Cupica), Golfo de Tribugá (G. Tri), Gorgona (Gorg), and Malpelo (Malp). Trophic groups: Herbivores-detritivores (HD), macroalgae-feeders (HM), sessile invertebrate feeders (IS), mobile invertebrate feeders (IM), planktivores (PK), piscivores (PS), and omnivores (OM).

overlooked in global assessments. Our results revealed that fish assemblages in seven location along the CPC (*i.e.,* Cabo Marzo, Cupica, Bahía Solano, Golfo de Tribugá, Cabo Corrientes, Gorgona, and Malpelo) exhibited a higher species richness in locations with some protection status. Conversely, fish density and biomass were higher in areas further from markets, including the geographically isolated Malpelo Island, situated 380 km from the coast. Further, we show that the proportion of fish families and trophic groups also varied across the CPC, and fish density and biomass per trophic group are influenced by market distance (*i.e.,* a human factor). Our results provide insights into fish assemblages in the CPC and contribute to advancing knowledge in this understudied region.

## Spatial variation of fish assemblage metrics

Fish assemblages across the Colombian Pacific Coast showed differences in species richness, density, and biomass. For instance, we observed a high species richness in Golfo de Tribugá, Cabo Corrientes, and Gorgona, which can be associated with structural complexity and variety of habitats in these locations. These locations are characterized by high complexity associated with the presence of rocky and coral reef areas, as well as abundant octocoral communities, some of them with high coral diversity for the Eastern Tropical Pacific region (*Zapata & Vargas-Ángel, 2003*; *Mejía-Quintero & Chasqui, 2020*). Similar results have been obtained in the Gulf of Papagayo in Costa Rica, Caribbean, and Indo-Pacific regions, where complex reefs support high species richness by providing refuges that reduce encounters between competitors and predators (*Dominici-arosemena et al., 2005*; *Hixon & Menge, 1991*; *Komyakova, Munday & Jones, 2013*). Furthermore, high complexity promotes the co-occurrence of species that are specialized in different reef habitats, such as vertical walls, horizontal reef matrices, and caves (*Depczynski & Bellwood, 2004*). Low fish species richness observed in Bahía Solano can be associated with low coral development, which results from a high sedimentation rate around this location (*Mejía-Quintero & Chasqui, 2020*). On the other hand, the low species richness observed in Malpelo likely results from the high isolation from the coast, which limits colonization, particularly of species with low dispersal capacity (*MacArthur & Wilson, 1967*; *Luiz et al., 2012*).

The high density and biomass observed in Malpelo result from different factors like island mass effects, which increase nutrients and energy available through upwelling, favoring a high density and large species (*Gove et al., 2016*). Furthermore, due to its geographical location in the center of the Eastern Tropical Pacific Marine Corridor, Malpelo receives large predators migrating among Cocos Island, Malpelo, and Galapagos in search of food, refugia, and cleaning services (*Edgar et al., 2011*; *Quimbayo et al., 2017a*). On the other hand, due to its high isolation (*i.e.,* 380 km from the mainland), Malpelo has a low human influence, resulting in a high concentration of top predators and large species that favor high fish biomass (*Sandin et al., 2008*; *Quimbayo et al., 2019*). Furthermore, Malpelo is a no-take marine protected area declared a UNESCO World Heritage Site, contributing to the high fish density and biomass observed on this remote island (*Quimbayo et al., 2017b*). The density and biomass observed in Cabo Corrientes and Cabo Marzo could be attributed to local oceanography conditions in the Panama Bight, such as vertical stratification of the

water column, which promotes upwelling and increases nutrient availability (*Rodríguez-Rubio, Schneider & Abarca del Rio, 2003*; *Giraldo et al., 2008b*). Lastly, we observed low fish biomass in Gorgona Island despite its conservation status (as Marine Protect Area) for 39 years. This result was unexpected since older MPAs are generally associated with a higher biomass of predators and large herbivores (*Aburto-Oropeza et al., 2011*; *Sandin et al., 2008*). One possible explanation is that lower biomass may be linked to the specific characteristics of the sites where data were collected, which were mainly in shallow areas with a high cover of branching corals. According to *Palacios & Zapata (2014)*, these areas tend to harbor small-size species and juvenile individuals, as the branching structure provides protection and food. However, this configuration with intertwined branches, creates a less complex habitat, thereby reducing the presence of large species, which are more abundant in deep and complex sites (*Dominici-Arosemena & Wolff, 2006*). Thus, future studies can explore other locations within Gorgona to determine whether fish biomass on this island is lower than found in other locations along the Colombian Pacific Coast.

## Spatial variation of trophic groups and fish families

The high species richness observed in the families Carangidae, Haemulidae, Labridae, and Pomacentridae also have been observed in other locations in the Eastern Tropical Pacific (*Dominici-Arosemena & Wolff, 2006*; *Arias-Godínez et al., 2019*). This pattern can be associated with a high number of species within these fish families, which are recognized as some the most speciose families globally (*Floeter et al., 2018*; *Rabosky et al., 2018*; *Morais, Ferreira & Floeter, 2017*). Further, many species within these families (*i.e.,* Carangidae) have a circumglobal distribution associated with the family's large body sizes and long pelagic larval durations (*Luiz et al., 2013*), which can contribute to the high species richness observed within this family. We observed that Acanthuridae and Haemulidae were families with higher density observed in all locations, which can be associated with their schooling behavior, forming medium and large schools in coastal waters (*Quimbayo et al., 2021*). Similar patterns have been observed in other locations in the Eastern Pacific, where these families have high relative contribution to the total observed abundance (*Dominici-arosemena et al., 2005*; *Alvarez-Filip, Reyes-Bonilla & Calderon-Aguilera, 2006*; *Dominici-Arosemena & Wolff, 2006*; *Arias-Godínez et al., 2019*). Finally, the high biomass observed in Ephippidae was unexpected, since this family is not usually identified as an important contributor to total biomass in the ETP (*Aburto-Oropeza et al., 2011*; *Quimbayo et al., 2017b*). This result may reflect the large body sizes and schooling behavior observed in species within Ephippidae (*Cordeiro et al., 2021*), as well as some external factors that affect other families through fishing activities or oceanography conditions that regulate food availability.

The high species richness, density, and biomass within piscivores in Malpelo support previous studies that showed a high concentration of a wide variety of top predators and mesopredators at this island (*Soler, Bessudo & Guzmán, 2013*; *Bessudo & Álvarez-León, 2014*; *Quimbayo et al., 2017b*). In contrast, the high species richness of sessile invertebrate feeders in Cabo Marzo and Golfo de Tribugá, as well as the high density of herbivores-detritivores in Cabo Marzo, can be associated with the high algal cover and diversity

of substrates in these locations (*Rincón-Díaz et al., 2020*; *Lizarazo Rodríguez et al., 2020*; *Mejía-Quintero & Chasqui, 2020*). Finally, the high biomass of omnivores in Cupica can be attributed to the limited presence of other trophic groups. This scarcity might facilitate resource partitioning among species, leading to an increased contribution of omnivorous species to the observed fish biomass. Similar patterns have been previously reported in oceanic islands (*Mendes et al., 2019*).

## Effect of human and conservation factors on fish assemblages

Our results showed that market distance, the number of fishermen, and protection status did not affect the total species richness per location, regardless of whether Malpelo was included or not. However, we observed a higher concentration of species richness per trophic group in locations within marine protected areas since these areas tend to experience lower human pressure (*Lester & Halpern, 2008*; *Cinner et al., 2013*). For instance, sites in Golfo de Tribugá, Cabo Corrientes, and Gorgona are located within protected areas where fishing activities are regulated, and some areas in these locations are designated as exclusive fishing zones (*Castellanos-Galindo & Zapata, 2018*; *Chasqui, 2020*). This division between different zones promotes sustainable fishing practices and favors the concentration of high species richness, including top predators and large species, as observed in other protected areas in the Eastern Tropical Pacific (*Aburto-Oropeza et al., 2011*; *Beita-Jiménez et al., 2019*). Other indirect factors that may influence species distribution across trophic groups along the Colombian Pacific Coast include isolation level, primary productivity, and water transparency (*Soler, Bessudo & Guzmán, 2013*; *Bessudo & Álvarez-León, 2014*; *Quimbayo et al., 2017b*; *Quimbayo et al., 2019*; *Dubuc et al., 2023*). For example, areas such as Cabo Marzo, Cupica, Bahía Solano, Golfo de Tribugá, Cabo Corrientes, and Gorgona are influenced by the upwelling in the Panama Bight and by various rivers that flow into the Colombian Pacific Ocean (*e.g.*, San Juan River and Patía-Sanquianga River delta; *Restrepo & Kjerfve, 2000*; *Giraldo, Rodríguez-Rubio & Zapata, 2008*), which increase nutrient levels in the water and consequently influence species richness.

We observed that market distance was the most important human driver that negatively influenced total biomass. These results support global trends where human populations nearest to reefs and other coastal habitats have a negative effect on fish assemblages (*Mora et al., 2011*; *Cinner et al., 2018*). In the regional context, this result was expected, given that the human populations along the Colombian Pacific Coast have high fish consumption, as highlighted in previous studies (*Castellanos-Galindo & Zapata, 2018*; *Herrón et al., 2018*; *Herrón et al., 2019*). For instance, several piscivores, such as *Scomberomorus sierra* (total fishing biomass at landing sites of the Colombian Pacific Coast based on *Herrón et al. (2019)*, approx. 76,580 kg), *Seriola rivoliana* (approx 66,524 kg), *Caranx sexfaciatus* (approx. 44,536 kg), and *Epinephelus quinquefasciatus* (approx. 22,128 kg) are prominent fishing targets due to their high commercial value in the region (*Castellanos-Galindo & Zapata, 2018*; *Herrón et al., 2018*). Other demersal species, especially snappers (Lutjanidae) or mobile invertebrate feeders, are also fished for local consumption (*Castellanos-Galindo & Zapata, 2018*). Despite this pattern, Colombia's Pacific Coast has the lowest fish consumption per capita per year in the Eastern Pacific region (*FAO, 2014*), which we

hypothesize contributes to lower human effects in this study compared to other areas around the world (*Pinca et al., 2012*).

## ACKNOWLEDGEMENTS

We acknowledge Colombia's National Natural Parks for study permits at Gorgona and Malpelo Islands. We also thank MacDonald C, two anonymous reviewers, and the editor for constructive feedback on earlier version of this article. This work is contribution number 1402 of INVEMAR.

### Funding

The authors received no funding for this work.

### Competing Interests

Juan P. Quimbayo is an Academic Editor for PeerJ and the leader of BioScales Lab, University of Miami.

### Author Contributions

- Juan P. Quimbayo conceived and designed the experiments, performed the experiments, analyzed the data, prepared figures and/or tables, authored or reviewed drafts of the article, and approved the final draft.
- Luis Chasqui conceived and designed the experiments, performed the experiments, authored or reviewed drafts of the article, and approved the final draft.
- Natalia Rincón-Díaz conceived and designed the experiments, performed the experiments, authored or reviewed drafts of the article, and approved the final draft.
- Adriana Alzate conceived and designed the experiments, performed the experiments, authored or reviewed drafts of the article, and approved the final draft.
- Fernando A. Zapata conceived and designed the experiments, performed the experiments, authored or reviewed drafts of the article, and approved the final draft.

### Data Availability

The data from Malpelo is available at Quimbayo et al. (2017). The data from the other locations are available in the Supplementary File.

Part of the data compiled from Gorgona Island is available at Alzate (2022) and at figshare: Alzate Adriana (2022). Alzate.Gorgona_ReefFishData.xlsx. figshare. Dataset. https://doi.org/10.6084/m9.figshare.19314110.v1.

Trait species data were compiled from Quimbayo et al. (2021) and are available at Zenodo: Juan Pablo Quimbayo, Fernanda Carolina da Silva, Thiago Costa Mendes, Débora Silva Ferrari, Samara Leopoldino Danielski, Mariana Gomes Bender, Valeriano Parravicini, Michel Kulbicki, & Sergio Ricardo Floeter. (2021). Life-history traits, geographical range and conservation aspects of reef fishes from the Atlantic and Eastern Pacific. Ecology. https://doi.org/10.5281/zenodo.4455016.

The R code that supports the findings of this study is available at GitHub and Zenodo:
- https://github.com/BioScalesLab/Colombian_Fish_Community.git.
- Quimbayo, J. P., Chasqui, L., Rincón-Díaz, N., Alzate, A., & Zapata, F. A. (2025). Human and conservation factors affect spatial variation of reef fish assemblages in Colombian Pacific reefs [Data set]. Zenodo. https://doi.org/10.5281/zenodo.15359238.

## Supplemental Information

Supplemental information for this article can be found online at http://dx.doi.org/10.7717/peerj.19482#supplemental-information.

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
