# Peer review of "Human and conservation factors affect spatial variation of reef fish assemblages in Colombian Pacific reefs"

_PeerJ, doi:10.7717/peerj.19482_

## Round 0.1 · original submission · Minor Revisions

Dear Dr. Quimbayo,

Your paper has been reviewed by two experts in the field. They agree that your research was well-executed and your manuscript provides relevant information for management strategies. They also provided suggestions that I hope you address before your manuscript is accepted for publication. Please make sure to acknowledge their valuable contribution to the revised version.

Thank you for your submission to PeerJ.

Reviewer 1 ·

Basic reporting

Well written, amply cited, professional quality figures and statistics.

Experimental design

Well replicated, explicitly described methods. Appropriate for the journal and audience, fills a critical research gap for an under-studied region of the world.

Validity of the findings

Conclusions are well stated although I would suggest reducing the scope of inference regarding environmental drivers of variation among sites in fish communities in the discussion, since these were not explicitly studied.

Annotated reviews are not available for download in order to protect the identity of reviewers who chose to remain anonymous.

Reviewer 2 ·

Basic reporting

The authors deserve commendation for their diligent work and for assembling this dataset over the years, particularly in a region that remains poorly understood. Overall, the manuscript maintains clarity and professionalism in its use of English. However, I've noted areas where clarity could be significantly enhanced (as detailed in the annotated PDF).

While the authors offer a thorough literature review, one crucial aspect remains lacking: the impact of marine protected areas (MPAs) on fish communities. Highly protected and actively managed MPAs are widely recognized as the most effective conservation tool for replenishing fish biomass and restoring entire ecosystems. Conversely, MPAs with minimal protection, permitting commercial and industrial activities, often yield negligible benefits and may not differ significantly from unprotected areas. The study region features a diverse array of conservation measures with varying levels of protection and implementation, a facet that the authors overlook, and a valuable opportunity to contribute to this growing body of research. This information is vital for a comprehensive understanding of their results.

The figures and tables are appropriately presented, and I commend the authors for including raw data and supplementary materials.

Experimental design

The research manuscript aligns well with the aims and scope of the journal. The research questions are well defined, relevant and address a significant knowledge gap in our field

However, the limited sample sizes concern me and I wonder about the robustness of the inferences within a location with only two sites (Golfo de Tribugá). Nonetheless, the authors have employed appropriate and robust statistical methods to maximize the utility of their data. To uphold the principles of scientific reproducibility, I encourage the authors to share not only their raw data but also the R scripts they used in their analysis. This transparency would facilitate the replication of their findings and further contribute to the advancement of scientific knowledge in our field.

One additional analysis I'd recommend relates to the use of length-weight parameters. I understand the authors used a combination of previously published estimates and those provided by Fishbase. But, I'd be very interested to know if the fish biomass estimates change substantially from using one source vs the other.

Lastly, the authors do not mention or control for seasonal differences in sampling. If this is not a concern, the authors need to justify why.

Validity of the findings

The conclusion of the paper is well-articulated, but I have a few suggestions to enhance the robustness of the analyses and the relevance of the conclusions:

1. Consider excluding the island of Malpelo from the analysis to avoid confusing readers. Malpelo's substantial biogeographical differences, distinct threats, and management histories suggest that its inclusion may not add significant value to the coastal-focused analyses. Furthermore, much more research has been published about Malpelo's fish compared to the coastal region of Colombia. Instead, highlight Malpelo in the discussion section while concentrating the analyses solely on the coastal sites, including Gorgona. This adjustment would strengthen statements such as the lack of correlation between distance from markets and protection level, given Malpelo's considerable distance from markets and its status as one of the most strictly protected areas in the country.

2. The study area presents interesting gradients in oceanography and geography that could potentially explain some observed patterns but have not been adequately addressed. For example, regions like Cabo Marzo and Cabo Corrientes, which exhibit some of the highest fish biomass, are distinct features along the coast characterized by strong currents, upwelling, and periods of increased productivity. I suggest exploring whether the main conclusions of the paper, such as the influence of distance to markets on fish assemblages and biomass, remain consistent when considering these factors.

3. I highly recommend that the authors give due attention to the nuanced and critical aspect of protection in their analysis. One suggestion is to use the MPA classification system outlined by Grorud-Covert (2023). The SSF Malpelo, PNN Gorgona, and the coastal DRMI exhibit varying levels of protection, and as such, we should anticipate distinct conservation outcomes for each. Lumping MPAs into a single "protection" category may explain the finding that species richness was fish biomass is higher in isolated sites but not in protected sites. The truly protected areas in the region (Gorgona and Malpelo) and also the most isolated.

4. An unreported finding of this study is that in all the extensive surveys done, not a single shark appears to have been observed. This is worrisome and speaks directly to the condition of these reefs. Despite the region being sparsely populated, local fishing appears to have diminished shark populations to virtually zero. In the absence of sharks, what do authors hypothesize has taken their place?


Lastly, and very importantly, the conclusion that fish biomass in the DRMI and PNN Gorgona is not different from that in unprotected sites is a significant finding that warrants much greater emphasis. This carries direct implications for management decisions in this part of Colombia.

Annotated reviews are not available for download in order to protect the identity of reviewers who chose to remain anonymous.

---

## Round 0.2 · Minor Revisions

Dear Dr. Quimbayo,

Thank you for your great effort in revising the manuscript in response to the reviewers’ suggestions. In my opinion, the paper has improved considerably.
Before publication, I kindly ask that you address a few minor revisions suggested by Reviewer 1 and myself. My comments are included in the attached file and line numbers correspond to the tracked changes version.

Once again, thank you for your hard work—and that of your co-authors. I look forward to receiving your revised manuscript soon.

Best regards,
Guilherme

Reviewer 1 ·

Basic reporting

Clear and well written, occasional typos and grammatical errors are highlighted in review document.

Experimental design

Well done, proper statistical approach and interpretation.

Validity of the findings

Novel study location in data-defficient region.

Annotated reviews are not available for download in order to protect the identity of reviewers who chose to remain anonymous.

---

## Round 0.3 · accepted · Accept

The authors have addressed all reviewers' and editor's comments.